# Maternal Vitamin D Deficiency Is a Risk Factor for Infants’ Epigenetic Gestational Age Acceleration at Birth in Japan: A Cohort Study

**DOI:** 10.3390/nu17020368

**Published:** 2025-01-20

**Authors:** Tomoko Kawai, Seung Chik Jwa, Kohei Ogawa, Hisako Tanaka, Saki Aoto, Hiromi Kamura, Naho Morisaki, Takeo Fujiwara, Kenichiro Hata

**Affiliations:** 1Division of Fetal Development, National Research Institute for Child Health and Development, Tokyo 157-8535, Japan; 2Department of Obstetrics and Gynecology, Jichi Medical University, Shimono 329-0498, Japan; jwa.seungchik@jichi.ac.jp; 3Center for Maternal-Fetal, Neonatal and Reproductive Medicine, National Center for Child Health and Development, Tokyo 157-8535, Japan; ogawa-k@ncchd.go.jp; 4Department of Social Medicine, National Center for Child Health and Development, Tokyo 157-8535, Japan; tanaka-hs@ncchd.go.jp (H.T.); morisaki-n@ncchd.go.jp (N.M.); 5Medical Genome Center, National Research Institute for Child Health and Development, Tokyo 157-8535, Japan; 6Department of Maternal-Fetal Biology, National Research Institute for Child Health and Development, Tokyo 157-8535, Japankhata@gunma-u.ac.jp (K.H.); 7Department of Public Health, Institute of Science Tokyo, Tokyo 113-8510, Japan; fujiwara.hlth@tmd.ac.jp; 8Department of Human Molecular Genetics, Graduate School of Medicine, Gunma University, Maebashi 371-8511, Japan

**Keywords:** maternal serum vitamin D, epigenetic clock, cord blood, DNA methylation

## Abstract

Background/Objectives: The DNA methylation of neonatal cord blood can be used to accurately estimate gestational age. This is known as epigenetic gestational age. The greater the difference between epigenetic and chronological gestational age, the greater the association with an inappropriate perinatal fetal environment and development. Maternal vitamin D deficiency is common in Japan. The aim of this study was to investigate the associations between maternal serum vitamin D levels and epigenetic gestational age acceleration at birth in Japan. Methods: The data were obtained from the hospital-based birth cohort study conducted at the National Center for Child Health and Development in Tokyo, Japan. Maternal blood was collected in the second trimester to measure the serum vitamin D concentration. Cord blood was collected at birth to measure serum vitamin D and to extract DNA. DNA methylation was assessed using an Illumina methylation EPIC array. Epigenetic gestational age was calculated using the “methylclock” R package. Linear regression analysis was performed to see associations. Results: Maternal serum vitamin D levels in the second trimester were negatively associated with epigenetic gestational age acceleration at birth when calculated by Bohlin’s method (regression coefficient [95% CI]: −0.022 [−0.039, −0.005], *n* = 157), which was still significant after considering infants’ sex (−0.022 [−0.039, −0.005]). Cord blood serum vitamin D levels were not associated with epigenetic age acceleration. Maternal age at delivery and birth height were associated in positive and negative ways with epigenetic gestational age acceleration, respectively (0.048 [0.012, 0.085] and −0.075 [−0.146, −0.003]). Conclusions: Maternal vitamin D deficiency was related to an infant’s epigenetic gestational age acceleration at birth. These findings suggest that the association between fetal development and maternal vitamin D levels may involve the fetal epigenetic regulation of the fetus.

## 1. Introduction

Vitamin D deficiency has been frequently observed in every generation worldwide. Pregnant women are no exception [1]. The prevalence of a low vitamin D status in pregnant women was especially high in Asia [2,3]. The extra-skeletal effects of vitamin D are still under debate, but symptoms such as respiratory infections, obesity and cancer mortality may be associated, especially in those who are deficient. Although the optimal concentration for bone health is also controversial, the Institute of Medicine (IOM), USA recommends a cut-off level of 20 ng/mL in its 2011 Dietary Reference Intakes. With regard to the upper limit of the concentration, 40 ng/mL has been set [4,5,6]. Maternal vitamin D intake during pregnancy or cord blood 25-hydroxyvitamin D (25[OH]D) is inversely associated with the risk of recurrent wheezing in children in the northeastern United States [7,8]. In a Greek cohort, maternal vitamin D deficiency at delivery was associated with low neonate birth weights [9]. In a multisite prospective cohort study in the Unites States, each 0.4 ng/mL increase in first-trimester 25(OH)D was associated with an increase in length-for-age z-scores but was not associated with weight or head circumference [10]. In addition, prenatal vitamin D deficiency is shown to program adipose tissue metabolism in off-spring in a sex-dependent way [11] or epigenetically program embryonic hematopoietic stem cells, resulting in adipose macrophage infiltration and type 2 diabetes in mice [12]. These results indicate that immune cells are epigenetically affected in utero by vitamin D deficiency.

Chemical modifications of Deoxyribonucleic acid (DNA) letters and histone proteins regulate gene expression without changing the DNA sequence. The presence or absence of these modifications is known as epigenetic changes. Detecting epigenetics in the genome helps to understand the biological states of cells. DNA methylation patterns in several tissues have been used to predict age accurately. This epigenetic age is recognized as a biological clock. The difference between epigenetic age and chronological age is referred to as epigenetic age acceleration. Epigenetic age acceleration increases adult mortality risk [13,14]. As we and others have confirmed, there is a correlation between DNA methylation levels at specific sites in the genome and gestational age [15,16,17,18]. Bohlin et al. and Knight et al. published that the weighted average of DNA methylation at their selected sites in the genome in cord blood can be used to estimate epigenetic gestational age in the same issue of the journal [17,18]. Epigenetic gestational age acceleration is recognized as a biomarker of physiological development [17,18,19,20]. In addition to cord blood cells, the pediatric buccal epigenetic clock has recently been developed to non-invasively predict epigenetic age acceleration in infants [21], which showed increased epigenetic age acceleration at term-equivalent age in very preterm infants with neonatal infection [22]. Long-lasting effects of neonatal environments are also detected in buccal DNA methylation. Epigenetic age acceleration based on buccal cells at age 30 to 35 years in men with extremely low birth weights was significantly more than in those with normal birth weights [23].

From these backgrounds, we aimed to investigate an association between maternal vitamin D levels in gestation and epigenetic gestational age acceleration at birth to assess whether maternal vitamin D deficiency influences fetal development. Vitamin D is classified based on the source. D2 is produced in plants and fungi. D3 is produced in animals, including humans. In human blood, vitamin D3 is much more abundant than vitamin D2. Vitamin D3 is formed from provitamin D3, the final intermediate in cholesterol biosynthesis in the body, which is converted in the skin to pre-vitamin D3 by exposure to ultraviolet radiation and then to vitamin D3 by body temperature. Vitamin D is hydroxylated in the liver to 25-hydroxy vitamin D (25(OH)D), which circulates stably in the blood. Eventually, the biologically active form of vitamin D is 1,25-dihydroxyvitamin D, made from 25(OH)D by further hydroxylation in the kidney. Different people have different backgrounds in these biosynthetic processes. So far, maternal vitamin D3 supplementation slows infants’ epigenetic gestational age at birth in a randomized controlled trial [24]. We considered that the blood 25(OH)D level measurement more accurately represented the nutritional status than information on the presence or absence of supplementation intake. In the present study, we measured serum 25(OH)D concentrations in maternal second trimester peripheral blood and cord blood and then assessed the genome-wide DNA methylation of cord blood cells to see the effects of gestational vitamin D on the epigenetic age acceleration. 

## 2. Materials and Methods

### 2.1. Participants

The participants of this study were recruited from a hospital-based birth study conducted at the National Center for Child Health and Development (NCCHD) in Tokyo, Japan, from 2010 to 2013. In the study, pregnant women were recruited during the first trimester of pregnancy. Their informed consent for participation and that of their newborns was obtained from them at the beginning of the study. Blood samples were collected from mothers in the second trimester between 24 to 28 gestational weeks. Cord blood samples were collected at delivery. Mother–child pairs were excluded from this study if the mothers smoked during pregnancy, were diagnosed with a pregnancy complication during pregnancy, had a pre-existing disease or took medications during pregnancy. In addition, we only included newborns born after 37 gestational weeks. Within this group, we excluded newborns who were large for their gestational age (>90th percentile) or small for their gestational age (<10th percentile). Because stillbirth and small or large birth weights have an influence on DNA methylation in cord blood cells [15,23], mother–child pairs meeting the above conditions were ranked by maternal serum vitamin D concentration and periodically selected from the top and bottom of the rankings so that the median maternal serum vitamin D concentration was 20 ng/mL, which was considered adequate by IOM 2011 [6].

### 2.2. Maternal and Newborn Assessments

Pre-pregnancy body mass index (BMI) was defined as pre-pregnancy weight (kg) divided by height (m^2^). These were reported in the written questionnaire asking about pre-pregnancy weight, height and smoking and drinking habits, which was given at the first visit to the hospital. Gestational weight gain was calculated by subtracting the above pre-pregnant weight from the weight at the latest prenatal checkup. Gestational age (GA) was estimated using a combined method based on the last menstrual period, first-trimester ultrasound and 20-week ultrasound. Birth weight SD score—namely, birth weight z-score for GA according to Japanese reference data—was calculated from GA, birth weight, infants’ sex and parity through a program provided by the Japanese Society for Pediatric Endocrinology (downloaded from following site on 10 December 2024; http://jspe.umin.jp/medical/keisan.html). Birth height SD score was calculated from GA, birth height and infant’s sex through the same program as well. Blood samples were stored at 4 °C until being assayed the next day. Maternal or cord blood serum 25(OH)D concentrations were measured by an external laboratory (LSI Medience, Itabashi, Tokyo, Japan) using a competitive protein-binding assay, as previously described [25,26]. The inter- and intra-assay coefficients of variation were ≤10%.

### 2.3. DNA Methylation Analysis Using Umbilical Cord Blood Samples

DNA was collected from whole cord blood cells. DNA Methylation data were obtained using the Infinium MethylationEPIC BeadChip array (Illumina, San Diego, CA, USA), as previously described [27,28]. Methylation data were obtained with the iScan system (Illumina) as idat files and then processed using the ENmix (v.1.30.3) package in R-4.1.3 [29]. The manifest file was annotated using “IlluminaHumanMethylationEPICanno.ilm10b4.hg19”. Beta values were calculated by the “mpreprocess” of ENmix.

### 2.4. Calculation of Gestational Age Acceleration

Bohlin et al. [17] and Knight et al. [18] showed that gestational age can be accurately estimated from the methylation levels of 96 and 148 probes in cord blood using the Infinium 450K Methylation Array. The R package ‘methylclock’ contains both algorithms developed by them for the DNA methylation gestational age (DNAmGA) estimation [30]. We processed it in R-4.3.3. As Chen et al. described [24], unlike the 450K bead array, the MethylationEPIC bead array does not contain the eight and six CpG probes that Bohlin and Knight extracted for the calculation of DNAmGA [17,18].

### 2.5. Statistical Analyses

Association tests were performed with linear regression with and without adjustment for the infant’s sex. Spearman correlation tests were performed to see the relationship between characteristics and epigenetic age acceleration.

## 3. Results

### 3.1. General Characteristics of the Study Population

The median serum 25(OH)D levels at mid-gestation of 157 participants was 20.5 ng/mL (Figure 1). In total, 76 of the 157 participants had 25(OH)D levels below 20 ng/mL, which the IOM defined as an adequate serum level. Data on gestational weeks at birth, birth weight, birth height and cord blood 25(OH)D levels were collected for the newborn’s information. For the parents, data were collected on maternal age, paternal age, pre-pregnancy BMI and gestational weight gain (Table 1).

### 3.2. Correlation of DNAmGA with Chronological Gestational Age

DNAmGAs were significantly correlated with chronological gestational age, even though our samples were collected at a limited gestational age, such as between 37 and 42 weeks. The correlation coefficient was better in Bohlin’s method (r = 0.71) than in Knight’s method (r = 0.48) (Figure 2). Bohlin’s method estimated older DNAmGAs than Knight’s method.

### 3.3. Gestational Age Acceleration at Birth Was Associated with Maternal Serum 25(OH)D Levels but Not with Cord Blood 25(OH)D

A linear regression analysis between DNAmGA age accelerations and each character regarding perinatal factors revealed significant associations with the ones estimated by only Bohlin’s method (Table 2). Maternal blood 25(OH)D levels in mid-gestation and birth height showed significant negative associations, with a regression coefficient of −0.022 (−0.039 to −0.005: 95% CI). There was no significant association between DNAmGA age accelerations and cord blood 25(OH)D levels. Also, maternal age at delivery and birth height showed significant positive and negative associations with DNAmGA age accelerations, with a regression coefficient of 0.049 (0.013 to 0.085: 95% CI) and −0.071 (−0.142 to −0.005: 95% CI), respectively (Table 2). These associations were still significant after adjustment for the infant’s sex (Table 3). However, gender differences were observed in correlations. Correlation coefficients between maternal blood 25(OH)D levels and DNAmGA age accelerations were −0.227 (*p* = 0.037) and −0.170 (*p* = 0.153) in male and female infants, respectively (Figure 3). Although maternal blood 25(OH)D levels and birth height were both negatively associated with DNAmGA age accelerations, they had no significant correlation (*p* = 0.248). There was no correlation between maternal blood 25(OH)D levels and gestational age at delivery (*p* = 0.409).

## 4. Discussion

Our results showed a stronger correlation between chronological age and DNAmGA with Bohlin’s method than with Knight’s method, as other cohort studies showed [19,31]. Furthermore, when measured in Bohlin’s method, some prenatal environments were significantly associated with DNAmGA age accelerations differently. A negative association between maternal 25(OH)D and gestational age acceleration in our study suggested that maternal 25(OH)D deficiency might accelerate fetal development improperly. In this study, we examined an association between serum 25(OH)D levels and the fetal epigenetic clock because several factors could regulate serum 25(OH)D levels [32]. For instance, the variant in GC vitamin D binding Protein, cytochrome P450 Family 27 Subfamily B Member 1 or Vitamin D Receptor gene was shown to be associated with differences in serum 25(OH)D levels in pregnant women, respectively [33]. Abdominal obesity and solar UV-B exposure reduced and increased serum vitamin D levels, respectively [34,35]. However, the congruence of the results with the previous report, in which the supplementary intake of 4000 IU/day vitamin D3 during pregnancy was negatively associated with the infant’s gestational age acceleration at birth, could indicate that the active intake of vitamin D3 during pregnancy affects fetal development via the upregulation of the intrinsic 25(OH)D concentration [24]. A prospective birth cohort study reported the longitudinal associations between DNAmGA and long-term obesity [36,37] or child blood pressure [38]. Although a systematic review showed no consistent evidence that vitamin D supplementation during pregnancy has clinically meaningful health benefits for infants and pregnant women [39], our results indicated that, including supplement intake, the effects of adequate vitamin D supply from mother to fetus could be observed in infants in the future through regulating epigenetic development, assessed by DNAmGA. At the molecular level, maternal serum 25(OH)D interacts with the vitamin D receptor (VDR), a transcription factor, in the placenta [40]. The secondary effects of vitamin D signaling through placental regulation may affect the fetal epigenome. Alternatively, maternal and neonatal circulating 25(OH)D concentrations were well correlated, suggesting that maternal serum 25(OH)D directly affects fetal cells as it arrives at the fetus [41]. In fetal blood cells, active vitamin D and the VDR complex could open up chromatin and regulate DNA methylation [42]. This may happen at the sites in the genome whose DNA methylation is used to estimate epigenetic gestational age.

Various maternal metabolites are associated with gestational age acceleration at birth. Higher maternal plasma homocysteine concentrations are positively associated with gestational age acceleration [43]. Higher homocysteine levels are caused by vitamin B12 deficiency [44], which indicates that maternal inadequate “One carbon metabolism” could affect fetal gestational age acceleration. In addition, the maternal plasma fatty acid pattern characterized by higher concentrations of n-3 polyunsaturated fatty acids was reported to be associated with accelerated epigenetic gestational aging [45]. These results suggest that inclusive maternal nutrition cooperatively programs DNAmGA. It is known that Vitamin D deficiency increases the risk of anemia [46,47]. Vitamin D suppresses hepcidin expression in cell levels [48,49] and is supposed to increase serum iron. In vivo, high-dose one-time oral vitamin D3 intake reduces the circulating hepcidin concentration in healthy adults [50]. Regarding pregnant women, positive relationships were observed in late pregnancy between plasma 25(OH)D and hepcidin and between plasma 25(OH)D and iron status, assessed by ferritin levels. However, vitamin D3 supplementation did not affect these two levels [51]. The discrepancy between serum 25(OH)D and vitamin D intake was also reported in women of reproductive age. The hemoglobin level in women of reproductive age was associated with serum 25(OH)D but not dietary vitamin D intake [52]. Different studies also showed that plasma 25(OH)D was positively associated with hemoglobin levels in all trimesters [53,54] and inversely associated with erythropoietin at both midgestation and delivery [55]. These results suggested that assessing plasma 25(OH)D levels tells more accurate causal relationships. Further investigation is needed to elucidate how the maternal plasma 25(OH)D status affects fetal DNAmGA, considering maternal anemia or iron metabolism.

Maternal age was positively associated with DNAmGA age acceleration in this study. This was partially consistent with the previous report, in which maternal age was associated with DNAmGA age acceleration of the offspring at birth but was limited to over 40 years [55]. The mean maternal age at delivery in our study is 37.5 ± 3.7. A different cohort study showed no association with maternal age at delivery, even after being classified by age as more than 35 years old. Meanwhile, the study showed a significant association between DNAmGA age acceleration and maternal parity [56]. Maternal physiological influences on the fetus cannot be derived only from maternal age. Parity and other physical characteristics must also be taken into account. Further studies are needed to elucidate the effects of maternal age. Regarding infant characteristics, several reports showed an association between DNAmGA age acceleration and birth weight. However, this study showed DNAmGA age acceleration was negatively associated with birth height. Maternal vitamin D status is reported to interact with fetal skeletal growth in utero [57]. The fact that birth height was associated in the same direction as maternal vitamin D levels may be related to bone development. Other measures, such as subcutaneous fat mass or neonatal bone densitometry, may show a more robust association with DNAmGA age acceleration. Shimpkin et al. showed that DNAmGA age acceleration at birth was associated with a higher fat mass on average across childhood in the cohort study that followed 1018 children from 0 to 17 years of age [37]. Further study is needed to reveal the associations between one’s birth height and epigenetic clock because the SD score of birth height did not associate with DNAmGA age acceleration.

Lastly, it is reported that the first-trimester maternal serum 25(OH)D, but not the one of the second trimester, is positively associated with fetal growth patterns between 16 and 42 gestational weeks [10]. Meanwhile, second-trimester maternal 25(OH)D concentrations were positively correlated with neonates’ birth weight, body length and head circumference at significant levels [58]. We collected maternal serum 25(OH)D levels only once during gestation for investigating the impact of initial programming on newborns. The survey period was the earliest point at which we were able to recruit most pregnant women in our institutional hospital. It is reported that maternal serum 25(OH)D levels gradually increased during pregnancy [10,59,60]. Francis et al. reported an average increase in maternal 25(OH)D levels of 1.6 nmol/L from 23–31 to 33–39 weeks’ gestation [60]. On an individual level, the percentage of body fat and the muscle mass may be related to the intra-individual changes in serum 25(OH)D levels during pregnancy [61]. Collecting the data for 25(OH)D levels at several time points during pregnancy could reveal whether the impact of maternal 25(OH)D has a critical gestational period regarding DNAmGA age acceleration.

## 5. Conclusions

We found an association between maternal mid-pregnancy vitamin D deficiency and neonates’ gestational age acceleration, as estimated by DNA methylation, in cord blood cells. Our result is consistent with a previous study, which showed that vitamin D supplement intake during pregnancy decreases gestational age acceleration. Our study could be limited by the sample size, collection period, potential selection bias or residual confounding. Future studies of the relationship between maternal vitamin D deficiency and neonates’ DNAmGA age acceleration could include larger sample sizes and racial/ethnically diverse samples. The longitudinal effects of maternal vitamin D and age acceleration at development stages are supposed to be considered.

## Figures and Tables

**Figure 1 nutrients-17-00368-f001:**
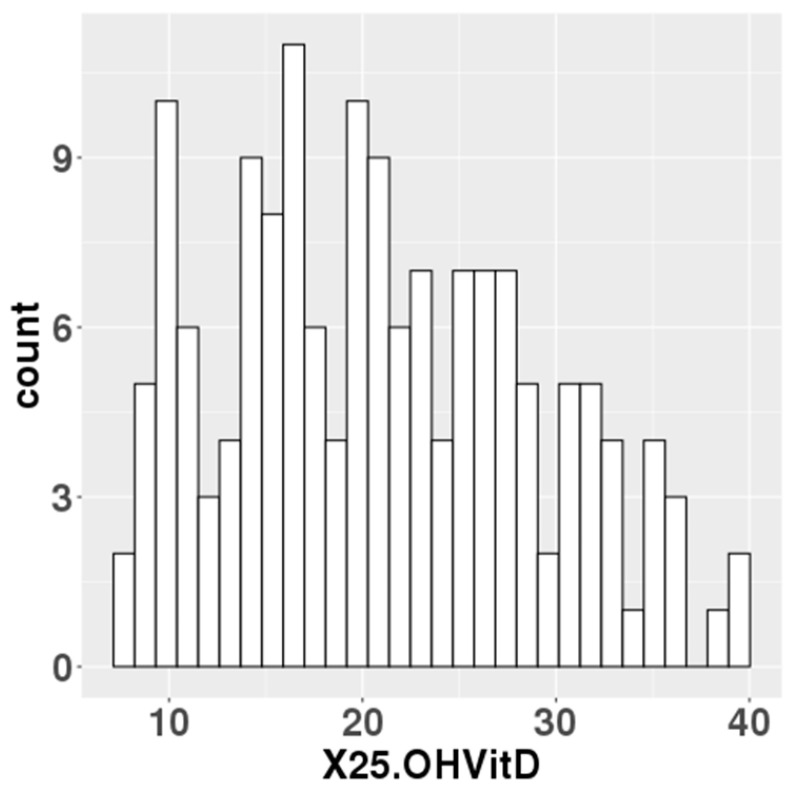
Maternal blood 25(OH)D levels in the study population.

**Figure 2 nutrients-17-00368-f002:**
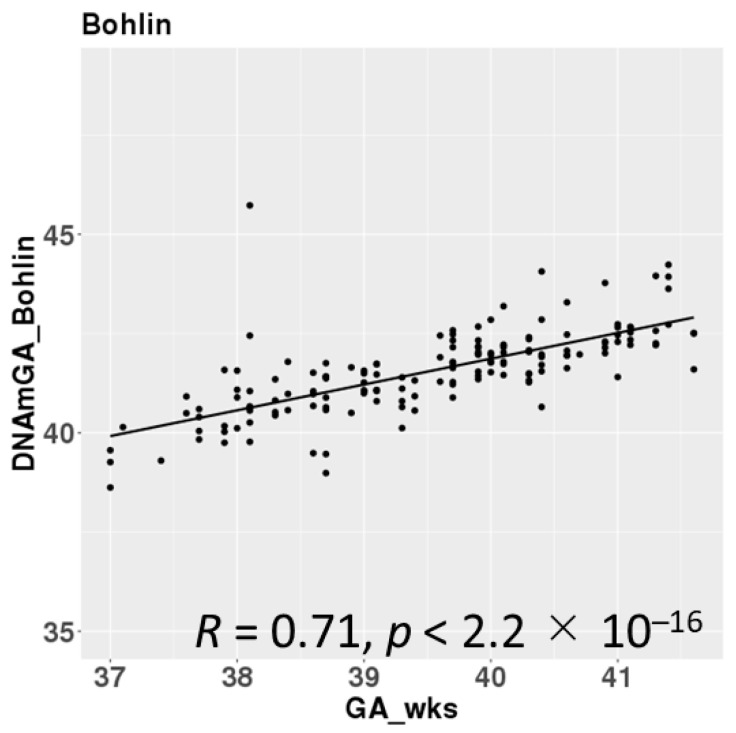
Correlations between chronological gestational age and DNAmGAs.

**Figure 3 nutrients-17-00368-f003:**
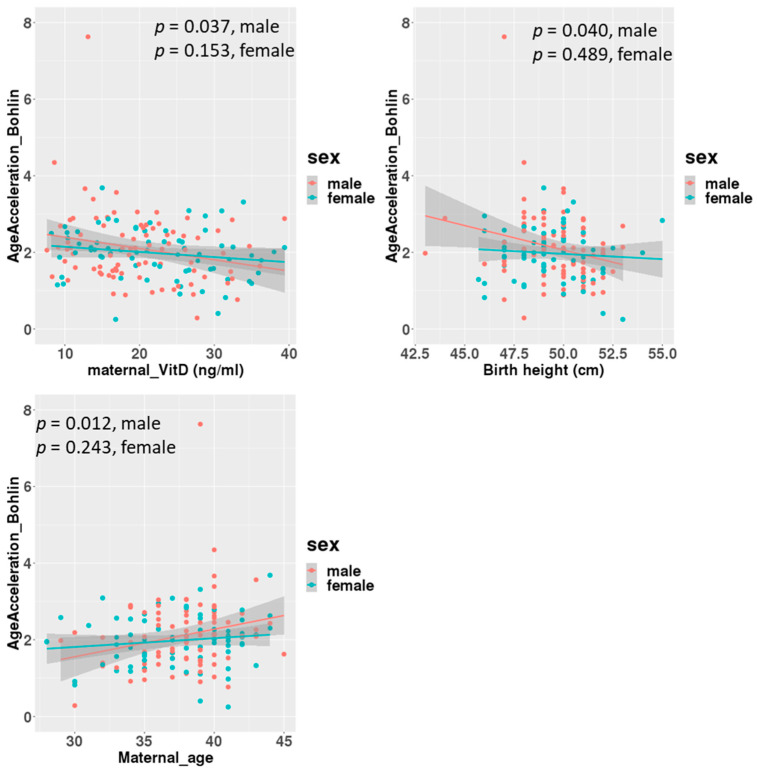
Correlations between age acceleration estimated by Bohlin’s method and maternal 25(OH)D levels, birth height and maternal age at delivery, adjusted for infants’ sex.

**Table 1 nutrients-17-00368-t001:** General characteristics of the participants (*n* = 157).

Parental Characteristics	
Age at delivery (years)	37.5 ± 3.7
Paternal age at newborn’s birth (years)	38.9 ± 8.2 (missing *n* = 15)
Pre-pregnant BMI (kg/m^2^)	19.97 ± 2.48
Gestational weight gain (kg)	10.64 ± 4.98
Maternal blood 25(OH)D (ng/mL)	21.02 ± 7.87
Newborn’s Characteristics	
Birth weight (g)	3054 ± 383
Birth height (cm)	49.5 ± 1.9
Gestational age at birth (weeks)	39.52 ± 1.15
Birth weight SD score	0.10 ± 1.02
Birth height SD score	0.26 ± 0.92
Cord blood 25(OH)D (ng/mL)	13.15 ± 4.39 (missing *n* = 3)

**Table 2 nutrients-17-00368-t002:** Associations of characteristics with AgeAcceleration of DNAmGA analyzed by simple linear regression. *: *p* < 0.05.

	AgeAcceleration_Bohlin		AgeAcceleration_Knight	
	Regression Coefficiency (95% CI)	*p* Value	Regression Coefficiency (95% CI)	*p* Value
Parental Characteristics				
Age at delivery (years)	0.049 (0.013, 0.085)	0.009 *	0.016 (−0.051, 0.083)	0.635
Paternal age at newborn’s birth (years)	0.003 (−0.015, 0.021)	0.748	−0.002 (−0.034, 0.030)	0.917
Pre-pregnant BMI (kg/m^2^)	−0.005 (−0.059, 0.049)	0.856	0.021 (−0.078, 0.119)	0.676
Gestational weight gain (kg)	0.002 (−0.026, 0.029)	0.913	0.021 (−0.028, 0.070)	0.398
Maternal blood 25(OH)D (ng/mL)	−0.022 (−0.039, −0.005)	0.010 *	−0.015 (−0.046, 0.016)	0.335
Newborn’s Characteristics				
Birth weight (g)	−0.0002 (−0.0006, 0.0001)	0.192	−0.0003 (−0.0009, 0.0004)	0.411
Birth height (cm)	−0.071 (−0.142, −0.005)	0.048 *	−0.105 (−0.234, 0.024)	0.109
Birth weight SD score	0.069 (−0.062, 0.200)	0.302	0.044 (−0.195, 0.284)	0.715
Birth height SD score	0.020 (−0.128, 0.167)	0.792	−0.081 (−0.348, 0.186)	0.551
Cord blood 25(OH)D (ng/mL)	−0.016 (−0.047, 0.015)	0.299	−0.011 (−0.068, 0.045)	0.695

**Table 3 nutrients-17-00368-t003:** Associations of characteristics with AgeAcceleration of DNAmGA adjusted for infant sex, analyzed by multiple linear regression. *: *p* < 0.05.

	AgeAcceleration_Bohlin		AgeAcceleration_Knight	
	Regression Coefficiency (95% CI)	*p* Value	Regression Coefficiency (95% CI)	*p* Value
Parental Characteristics				
Age at delivery (years)	0.048 (0.012, 0.085)	0.009 *	0.016 (−0.050, 0.084)	0.630
Paternal age at newborn’s birth (years)	0.002 (−0.016, 0.020)	0.852	−0.001 (−0.034, 0.031)	0.928
Pre-pregnant BMI (kg/m^2^)	−0.004 (−0.0058, 0.051)	0.895	0.019 (−0.080, 0.119)	0.698
Gestational weight gain (kg)	0.002 (−0.025, 0.029)	0.899	0.021 (−0.028, 0.070)	0.405
Maternal blood 25(OH)D (ng/mL)	−0.022 (−0.039, −0.005)	0.013 *	−0.017 (−0.048, 0.015)	0.299
Newborn’s Characteristics				
Birth weight (g)	−0.0002 (−0.0006, 0.0001)	0.170	−0.0003 (−0.0009, 0.0004)	0.433
Birth height (cm)	−0.075 (−0.146, −0.003)	0.040 *	−0.103 (−0.233, 0.027)	0.118
Birth weight SD score	0.072 (−0.060, 0.204)	0.285	0.042 (−0.199, 0.282)	0.733
Birth height SD score	0.014 (−0.135, 0.162)	0.858	−0.075 (−0.344, 0.195)	0.585
Cord blood 25(OH)D (ng/mL)	−0.019 (−0.050, 0.013)	0.238	−0.010 (−0.067, 0.048)	0.740

## Data Availability

The data are not publicly available due to privacy. The data that support the findings of this study are available from the corresponding author (T.K.), upon reasonable request.

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
