# Peer review of "Maternal Vitamin D Deficiency Is a Risk Factor for Infants’ Epigenetic Gestational Age Acceleration at Birth in Japan: A Cohort Study"

_nutrients, 2025, doi:10.3390/nu17020368_

Round 1

Reviewer 1 Report

Comments and Suggestions for Authors

I would like to thank you for granting me the opportunity to comment on this paper.

The MS is on the correlation between Maternal vitamin D deficiency and fetal epigenetic gestational age acceleration at birth in a Japanese cohort. The topic is interesting and there is a paucity in this research field. The literature search did not reveal any published article in this interesting and important research topic. The topic would be interesting for the readers of the Nutrition, and I would like to suggest the MS for publication, however, after considerable changes.

Abstract:

The introductory sentences (lines 25-29) are quite lyric and poetic instead of being scientific. I advise the authors to shorten this section and write more specifically about the scientific aim. The first four sentences should be rephrased. Moreover, it is unclear for a laic person what is the MS about. The authors just name infants' epigenetic gestational age acceleration, but readers like me have never heard of it and should be explained in a more common language.

Orthographic failure: Infants sex -> infants' sex in line 39

I think that the conclusion expressing that maternal vitamin D control the epigenetic program of the fetus is too speculative. The correlation between the two phenomena does not lead to the conclusion directly that maternal vitamin D determine the epigenetic gestational age acceleration of the fetus, since linear regression is a statistical probe that does not indicate a causative relationship.

Introduction

The first two sentences are the same as in the Abstract. Repetition should be excluded.

Line 54: English is not correct – 'to sex-dependently program adipose tissue metabolism in offspring' -> to program adipose tissue metabolism in offspring in a sex-dependent way.

Line 58: DNA should be written out

The authors should define more simply and comprehensively what does epigenetic age acceleration and what is the meaning of the fact that how DNA methylation show the physiological development. Is epigenetic age acceleration is a specific pattern of DNA methylation?

Line 80: So far, Maternal -> So far, maternal …

Materials and methods

Line 95: The authors state that they collected the maternal blood samples between the 24th and 28th weeks of gestation, but this time interval belongs to the third trimester.

The authors should specify which pregnancy complications and pre-existing diseases served as exclusion criteria (lines 96-97).

Was there any maternal drug use (for example Levaxin or other) that was ground for exclusion?

Line 98-100 are not correct in English. Only premature births between 28 and 37 weeks were included in the study or they were excluded from the study? Those newborns who were large for gestational age (>90th percentile) or small for gestational age (<10th percentile) were also excluded from the study? Of course, stillbirths were also left out of the study?

Line 101: What does intrinsic vitamin D level mean? A more explanation is needed for how the random selection happened if the authors chose the samples from a pool.

Line 102: What is IOM for? The sentence has to have reference.

Line 105: BMI should be written out.

Line 106: The authors are reporting on a questionnaire but they did not explain what kind of questionnaire is about? When was it distributed?

Discussion

The authors should discuss how stable is the maternal 25(OH)D level in the mid trimester. Do the authors think that the maternal vitamin D level is stable and unchanged throughout the second and third trimester? Why the authors chose to take a maternal serum sample for vitamin D level evaluation in the second trimester?

Line 126-130. The authors should interpret the DNA methylation gestational age as a more profound way. What is Bohlin and Knight methods?

Table 2 and 3 are not self-explanatory and it is not shown what kind of statistical methods are presented in the tables. Are they Spearman correlation coefficients of multivariate linear regression coefficients? The coefficients are otherwise rather low, but some of them are significant.

How can be explained the opposite correlation of birth weight and birth height with the DNAmGA?

Discussion section

The interpretation of age acceleration markers and their correlations to the clinical factors and their meanings would be beneficial. 

Reviewer 2 Report

Comments and Suggestions for Authors

This is an interesting study. However: 1) It should be mentioned that the optimal serum 25(OH)D concentration for certain conditions (as in skeletal health) is controversial. 2) An upper limit of normal 25(OH)D concentration should be mentioned too (40 ng/mL). 3) Some important references are missing (Example: Kokkinari, A., et al. (2024) Are Maternal Vitamin D (25(OH)D) Levels a Predisposing Risk Factor for Neonatal Growth? A Cross-Sectional Study. Clin Pract. 14, 265-279.

Reviewer 3 Report

Comments and Suggestions for Authors

The manuscript is relevant, however, I have some questions for the authors:

1. Line 58: the phrase "epigenetics are very informative biological markers" might lead to misinterpretation. Technically, epigenetics is the field of study. I suggest the authors reformulate the sentence.

2. Lines 72-73: Vitamin D is not "commonly referred to" as D2 and D3; it is classified based on the source (animal or vegetal). Please correct it.

3. The authors mentioned one of the exclusion criteria was maternal previous disease. What about medications used during pregnancy?

4. Line 128-129: Please explain what the missing probes in the Epic assay are.

5. Please include the P-Values in Figure 3.

6. Minor point: there is no need to include a subsection for figures and tables.

7. Please include one or two sentences explaining the molecular mechanisms of vitamin D's association with the epigenetic clock in the discussion.

Comments on the Quality of English Language

I have some doubts in some sentences, as I pointed out in the Comments and Suggestions section.

Round 2

Reviewer 3 Report

Comments and Suggestions for Authors

The authors performed good alterations in the manuscript and clarified my doubts.